

# Tumor mutational profile of triple negative breast cancer patients in Thailand revealed distinctive genetic alteration in chromatin remodeling gene

Suvimol Niyomnaitham[1], Napa Parinyanitikul[2],
Ekkapong Roothumnong[3,4], Worapoj Jinda[4,5], Norasate Samarnthai[6],
Taywin Atikankul[7], Bhoom Suktitipat[4,8,9], Wanna Thongnoppakhun[4,5],
Chanin Limwongse[3,4,5] and Manop Pithukpakorn[3,4]

[1] Department of Pharmacology, Faculty of Medicine Siriraj Hospital, Mahidol University, Bangkok, Thailand
[2] Department of Medicine, Faculty of Medicine, Chulalongkorn University, Bangkok, Thailand
[3] Department of Medicine, Faculty of Medicine Siriraj Hospital, Mahidol University, Bangkok, Thailand
[4] Siriraj Center of Research Excellence in Precision Medicine, Faculty of Medicine Siriraj Hospital, Mahidol University, Bangkok, Thailand
[5] Research Division, Faculty of Medicine Siriraj Hospital, Mahidol University, Bangkok, Thailand
[6] Department of Pathology, Faculty of Medicine Siriraj Hospital, Mahidol University, Bangkok, Thailand
[7] Department of Pathology, Queen Savang Vadhana Memorial Hospital, Thai Red Cross Society, Chonburi, Thailand
[8] Department of Biochemistry, Faculty of Medicine Siriraj Hospital, Mahidol University, Bangkok, Thailand
[9] Integrative Computational Bioscience Center, Mahidol University, Bangkok, Thailand

Corresponding author
Manop Pithukpakorn,
manop.pit@mahidol.ac.th

## ABSTRACT

**Background:** Triple negative breast cancer (TNBC) is a breast cancer subtype characterized by absence of both hormonal receptors and human epithelial growth factor receptor 2 (HER2). TNBC accounts for 15–20% of breast cancer. TNBC is associated with more aggressive disease and worse clinical outcome. Though the underlying mechanism of TNBC is currently unclear, the heterogeneity of clinical characteristics in various population may relate to the difference in tumor mutational profile. There were studies on TNBC gene mutations in various ethnic groups but the tumor genome data on Thai TNBC patients is currently unknown. This study aims to investigate mutational profile of Thai TNBC.
**Methods:** The patients were Thai individuals who were diagnosed with primary breast carcinoma between 2014 and 2017. All surgically removed primary tumor tissues were carefully examined by pathologists and archived as formalin-fixed paraffin-embedded tumor. TNBC was defined by absence of hormonal receptors and HER2 by immunohistochemistry. Genomic DNA was extracted, enriched and sequenced of all exomes on the Illumina HiSeq. Genomic data were then processed through bioinformatics platform to identify genomic alterations and tumor mutational burden.
**Results:** A total of 116 TNBC patients were recruited. Genomic analysis of TNBC samples identified 81,460 variants, of which 5,906 variants were in cancer-associated genes. The result showed that Thai TNBC has higher tumor mutation burden than

previously reported data. The most frequently mutated cancer-associated gene was *TP53* similar to other TNBC cohorts. Meanwhile *KMT2C* was found to be more commonly mutated in Thai TNBC than previous studies. Mutational profile of Thai TNBC patients also revealed difference in many frequently mutated genes when compared to other Western TNBC cohorts.

**Conclusion:** This result supported that TNBC breast cancer patients from various ethnic background showed diverse genome alteration pattern. Although *TP53* is the most commonly mutated gene across all cohorts, Thai TNBC showed different gene mutation frequencies, especially in *KMT2C*. In particular, the cancer gene mutations are more prevalent in Thai TNBC patients. This result provides important insight on diverse underlying genetic and epigenetic mechanisms of TNBC that could translate to a new treatment strategy for patients with this disease.

# INTRODUCTION

Triple negative breast cancer (TNBC) accounts for approximately 15–20% of breast cancer (*Blows et al., 2010*). Breast cancer patients with TNBC are not eligible for effective selective hormonal modulator or anti-HER2 treatments because of the absence of both hormonal and growth factor receptor overexpression. Chemotherapy was therefore the only available treatment of patients with TNBC. Women with TNBC displayed a clinical aggressiveness and high risk of metastasis. TNBC has also been shown to be associated with the poorer prognosis and reduced 5-year survival than other breast cancer subtypes (*Malorni et al., 2012*). Several studies showed the substantial racial variations of clinical behavior and prevalence of TNBC, likely owing to a heterogeneous nature of the disease. African descents are more often to present with TNBC, higher histologic grade and more aggressive breast tumors than whites (*Chen & Li, 2015*). Both Hispanic and African women tend to be diagnosed in more advanced stage (*Banegas & Li, 2012*). Studies in Asian populations demonstrated that 11% of breast cancer patients in Singapore had TNBC while this subtype accounted for 19% of Korean breast cancer patients (*Rhee et al., 2008*; *Thike et al., 2009*). The heterogeneity of TNBC on the clinical presentation as well as histologic differences may relate to variation in genetic background. The genomic profiles of TNBC in African Americans patients have been studied (*Ademuyiwa et al., 2017*; *Huo et al., 2017*). This study addressed the lack of data on the genomic profiles of TNBC in Thai and Asian population and investigated racial differences in the genetic landscape of breast cancer that could potentially identify targets suitable for specific population.

# METHODS
## Study population
The study protocol was approved by the Siriraj and King Chulalongkorn Memorial Hospital Institutional Review Boards (Protocol No. 175/2559 and 642/2557). The study

was conducted according to the Good Clinical Practice and the Declaration of Helsinki. All participants provided written informed consent. One hundred and sixteen Thai patients who were diagnosed with primary TNBC and treated at both hospitals between 2014 and 2017 were included. All 116 patients were female. The average age at diagnosis was 56.47 ± 11.90 years (±SD) with an age range between 25 and 79 years. The BMI was 26.28 ± 5.89 kg/m2. Majority of the patients (87%) were categorized as early stage breast cancer (20% as stage I and 67% as stage IIa). Fifty-five patients had follow-up period up to 3 years. There were six clinical relapses within the 3-years follow-up period; four cases were in stage IIa and two cases were in stage IV. One patient died during follow-up period whose cause of death did not appear to be cancer-related.

Primary tumor tissues and lymph nodes were surgically removed as a standard treatment at Siriraj and King Chulalongkorn Memorial Hospitals and examined by board-certified pathologists. The tissues were dissected for histological diagnosis and immunohistochemistry staining then archived as FFPE tumor block. TNBC subtype was defined by absence of estrogen receptor (ER), progesterone receptor (PR) and human epithelial growth factor receptor 2 (HER2) by immunohistochemistry staining with appropriate positive control. No amplification of *HER2* was also confirmed by in-situ hybridization. Independent pathologists determined only the samples of primary TNBC tumors, which had more than 50% tumor content after dissection, to be analyzed in this study. The FFPE primary tumors were sectioned into 10 micrometers using a new blade and preserved in 1.5 mL Eppendorf tubes. Blade was changed for every tissue block to prevent the contamination of DNA.

## Tumor genome sequencing and variant calls

The genomic DNA (gDNA) was extracted using Qiagen DNeasy DNA Isolation Kit (Hilden, Germany). FFPE gDNA (50–150 ng) was converted into libraries and enriched for whole exome sequencing using Agilent's SureSelect Human All Exon V5 + UTR Sample Prep kit. Sequencing was performed on the Illumina HiSeq 2500/4000 platform with average 300× sequencing depth. Genomic data were then processed through bioinformatics platform and knowledge base (Wuxi NextCODE Genomics, Boston, MA, USA) to identify genomic alterations including single nucleotide polymorphisms/substitutions (SNPs) and small insertions/deletions (indels). A threshold of 5% allelic fraction was used for SNPs and indels. Any variants presented at allele frequency above 1% in dbSNP, 1,000 Genomes and ExAC databases were removed. To assess somatic status of mutations in a tumor-only setting, we used both MuTect2 (Broad Institute, Boston, MA, USA) and VarScan2 (Washington University, St. Louis, MO, USA) on the aligned sequence data to determine somatic variants. All variants are further annotated with the extensive pipeline including COSMIC data annotations. Additional annotation includes HGMD Professional, ClinVar, OMIM and multiple missense functional predictors including polyphen2, SIFT, LRT, MutationTaster, MutationAssessor and CADD. The effect of the sequence variants on all protein-coding genes in the RefSeq database was further annotated using the Variant Effect Predictor, which predicts the consequence of each sequence variant on all neighboring

RefSeq genes based on a set of 35 consequence terms defined by the Sequence Ontology (*McLaren et al., 2016*). Only variants predicted to cause strong and moderate alteration on gene functions, such as stop gained/lost variants, frameshift, indels, donor/acceptor splice variants, initiator codon variants, missense variants, in-frame indels and splice region variants were selected for analysis. Finally, the results were manually reviewed by molecular geneticists. These analysis methods applied to the cancer genome atlas (TCGA) tumor variants and expression data have distinguished different molecular or histologic subtypes of breast cancer at over 97% accuracy.

## Data comparison with other breast cancer cohorts

To compare frequency of cancer gene alterations between this study and previously published data, breast cancer mutation data from TCGA, METABRIC and French cohorts were downloaded from the cBioPortal for Cancer Genomics (http://www.cbioportal.org). TCGA cohort consisted of cancer genome data form primary breast cancer patients in the United States (*Cancer Genome Atlas Network, 2012*). METABRIC cohort data were collected from primary breast cancer patients in the United Kingdom and Canada (*Pereira et al., 2016*). French cohort data were primarily from the patients with metastatic breast cancers from four different prospective trials in France (*Lefebvre et al., 2016*). Cancer gene mutation data from TCGA database were selected only from breast invasive ductal carcinoma that was classified as PAM50 basal subtype, which is closely related to triple negative subtype of breast cancer, while data from METABRIC and French cohorts were selected from samples with negative ER, PR and HER2, similar to this study. The comparison of mutation frequency of each cancer genes was trimmed to 173 genes due to data limitation from 173-gene panel in METABRIC cohort.

## Statistical analysis and data visualization

Genomic characteristics were compared across cohorts using one-way analysis of variance for continuous variables. The prevalence of somatic mutations was compared across cohorts. Descriptive statistics was used to show the average age and BMI of population. Statistical analyses were conducted using SAS software (version 9.4; SAS Institute, Cary, NC, USA). A two-tailed $p$ value less than 0.05 was considered significant. Landscape of co-occurrence and mutual exclusion of cancer gene mutations was generated with OncoPrinter (version 1.0.1; cBioPortal for Cancer Genomics, New York, NY, USA). Mutational spectrum of three most commonly mutated genes in lollipop plot format was generated with MutationMapper (version 1.0.1; cBioPortal for Cancer Genomics, New York, NY, USA) (*Gao et al., 2013*).

# RESULTS

## Comparison of gene alterations between Thai TNBC and other TNBC cohorts

A total of 1,088,237 variants were detected from the whole exome sequencing data of 116 patients. Variant filtering with MuTect2 and VarScan2 identified 81,460 somatic variants that passed data analysis algorithms. When all 969 known or potential

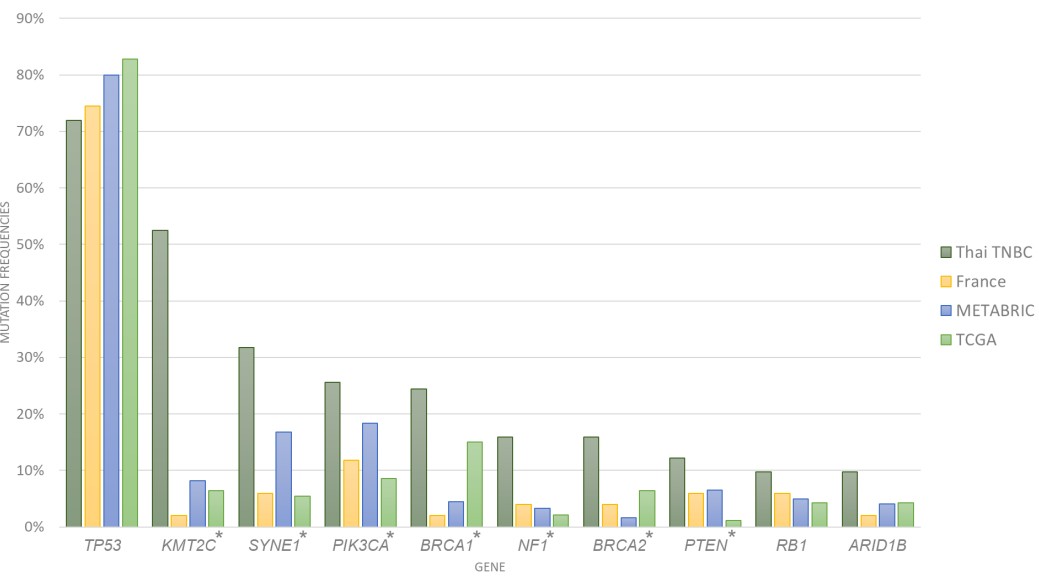

**Figure 1 Somatic mutation frequencies among four TNBC cohorts.** Bar chart showed 10 most commonly mutated in Thai TNBC compared to TCGA, METABRIC and French cohorts. * indicated the difference was statistically significant.

**Table 1 Frequencies of ten most commonly mutated genes in Thai TNBC compared to other cohorts.**

| Gene | Thai TNBC ($n$ = 116) (%) | TCGA ($n$ = 93) (%) | METABRIC ($n$ = 245) (%) | French ($n$ = 51) (%) | Pearson chi square ($p$ value) |
|------|---------------------------|---------------------|--------------------------|-----------------------|-------------------------------|
| TP53 | 75.86 | 82.80 | 80.00 | 74.51 | 0.519 |
| KMT2C | 57.76 | 6.45 | 8.16 | 1.96 | <0.001 |
| SYNE1 | 31.71 | 5.38 | 16.73 | 5.88 | <0.001 |
| PIK3CA | 23.28 | 8.60 | 18.37 | 11.76 | 0.027 |
| BRCA1 | 21.55 | 15.05 | 4.49 | 1.96 | <0.001 |
| BRCA2 | 18.10 | 6.45 | 1.63 | 3.92 | <0.001 |
| NF1 | 14.66 | 2.15 | 3.27 | 3.92 | <0.001 |
| PTEN | 11.21 | 1.08 | 6.53 | 5.88 | 0.033 |
| RB1 | 10.34 | 4.30 | 4.90 | 5.88 | 0.19 |
| ARID1B | 6.90 | 4.30 | 4.08 | 1.96 | 0.495 |

**Note:**
  $n$ indicated number of patients in each cohort.

cancer-associated genes were examined, only 5,906 somatic variants were identified. We found an average of 222 variants per sample (range 104–388 variants) with an average of 20 altered cancer genes (range 5–35 genes). The 10 most commonly mutated genes were *TP53, KMT2C, SYNE1, PIK3CA, BRCA1, NF1, BRCA2, PTEN, RB1* and *ARID1B* (Fig. 1; Tables 1 and 2). Compared to the cancer gene mutation frequencies from TCGA, METABRIC and French cohorts, Thai TNBC had significantly higher mutation frequencies in *KMT2C, SYNE1, PIK3CA, NF1, PTEN, BRCA1* and *BRCA2*. On the contrary, significant differences were observed only in *SYNE1* and *BRCA1* among TCGA, METABRIC and French cohorts.
**Table 2 Frequencies of mutated genes among three published cohorts.**

| Gene | TCGA ($n$ = 93) (%) | METABRIC ($n$ = 245) (%) | French ($n$ = 51) (%) | Pearson chi square ($p$ value) |
|------|------|------|------|------|
| TP53 | 82.80 | 80.00 | 74.51 | 0.494 |
| KMT2C | 6.45 | 8.16 | 1.96 | 0.278 |
| SYNE1 | 5.38 | 16.73 | 5.88 | 0.006 |
| PIK3CA | 8.60 | 18.37 | 11.76 | 0.063 |
| BRCA1 | 15.05 | 4.49 | 1.96 | 0.001 |
| BRCA2 | 6.45 | 1.63 | 3.92 | 0.068 |
| NF1 | 2.15 | 3.27 | 3.92 | 0.812 |
| PTEN | 1.08 | 6.53 | 5.88 | 0.123 |
| RB1 | 4.30 | 4.90 | 5.88 | 0.915 |
| ARID1B | 4.30 | 4.08 | 1.96 | 0.749 |

**Note:**
$n$ indicated number of patients in each cohort.

## Driver gene analysis

We performed analysis of the filtered cancer genes to identify potential genes of interest. We first examined for genes mutated in multiple samples and found that 14 genes were mutated in at least 10% of the samples. SYNE1 was excluded from analysis due to unclear role in cancer. TP53 was the most frequently mutated gene with variants found in 88 samples. KMT2C, which encodes for the lysine methyltransferase mixed-lineage leukemia (MLL3), was the next most frequently mutated gene with mutations in 67 samples. The third most frequently mutated gene was PIK3CA, which was shared by 27 samples. Together, 111 of the 116 samples had an alteration in at least TP53, KMT2C or PIK3CA (Fig. 2). The most commonly recurring mutation was the PIK3CA H1074R mutation, which was found in the majority of the PIK3CA mutant samples (16 out of 27 samples; 59%). Within these genes, we also found a number of recurrent variants including the previously reported mutations in KMT2C S338L (21 samples; 31%), TP53 R175H (10 samples; 11%), R248Q (five samples; 6%), Y220C (four samples; 5%) and R273H (four samples; 5%) and PIK3CA K733R (six samples; 22%) and E545K (four samples; 15%) (Fig. 3).

## TP53 mutation status and co-occurring mutations

As previously shown, TP53 was the most commonly mutated gene in the cohort with 88 of the 116 samples (76%) containing a mutation. To identify roles of TP53 and its association with the other two most commonly mutated genes; KMT2C and PIK3CA, we subdivided the cohort as TP53 wild-type and TP53 mutant groups. In the TP53 wild-type samples ($n$ = 28), the PI3K pathway appeared to be a predominant driver with 19 samples (68%) containing a mutation in either the PIK3CA gene or PI3K pathway members, including PIK3C2B, PIK3CG and PTEN. In the TP53 mutant group, 67 samples (76%) had one or more mutations in genes encoding chromatin remodeling proteins, including ATRX, DNMT3A and KMT2C, which have been reported to be

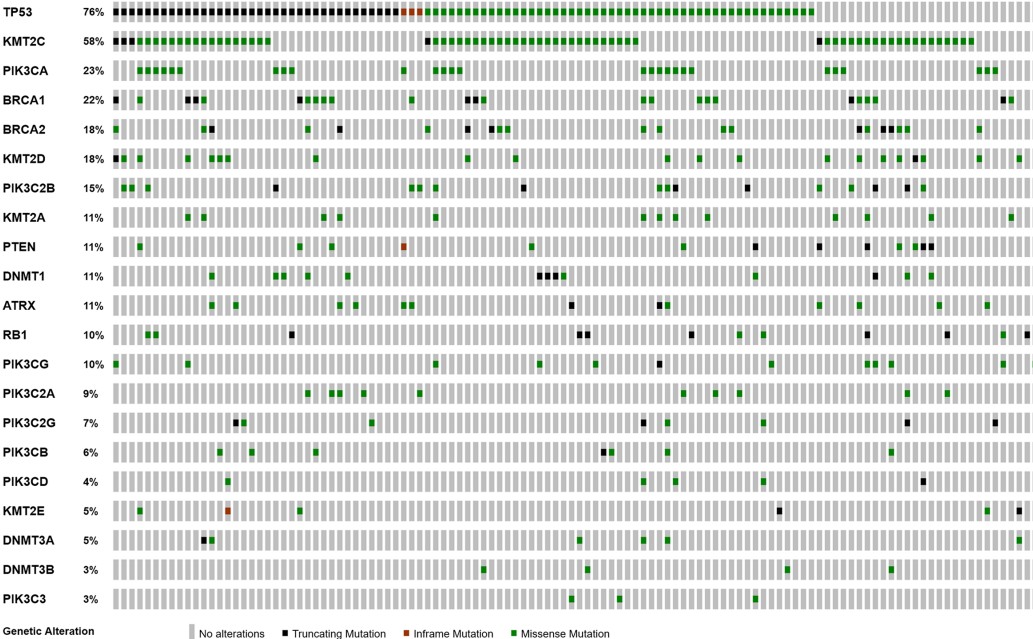

**Figure 2 Landscape of co-occurrence and mutual exclusion of cancer gene mutations in Thai TNBC.**

involved in cancer. This mutation co-occurrence may suggest a complex interplay in TNBC. However, there was no association between *TP53, PIK3CA* or *KMT2C* mutation status and cancer staging.

## Tumor mutation burden

One of the biological hallmarks in cancer is genome instability. Genome alterations, which either involve in carcinogenesis or occur as a result of widespread genome instability, can create neoantigens and trigger immune response. We used the variant data to calculate the tumor mutation burden for each sample and employed the same filtering scheme accounted for all variants including both nonsynonymous and synonymous calls. Tumor mutation burden was defined as the number of somatic base substitutions, and indels per megabase of coding genome sequence examined. Synonymous mutations are counted in order to reduce sampling noise. Though majority of synonymous mutations are not likely to cause tumor immunogenicity, they may reflect mutational processes that will also result in nonsynonymous mutations and neoantigens elsewhere in the genome (*Diederichs et al., 2016*; *Supek et al., 2014*). We found that Thai TNBC has an average tumor mutation 7.3 variants per megabase (95% CI [6.9–7.6]), which consists of nonsynonymous mutation 3.9 variants per megabase (95% CI [3.7–4.1]) and synonymous mutation 3.4 variants per megabase (95% CI [3.3–3.6]). This data showed significantly higher tumor mutation burden than median value (3.6 variants per megabase) found in invasive ductal breast carcinoma cohort (*Chalmers et al., 2017*).

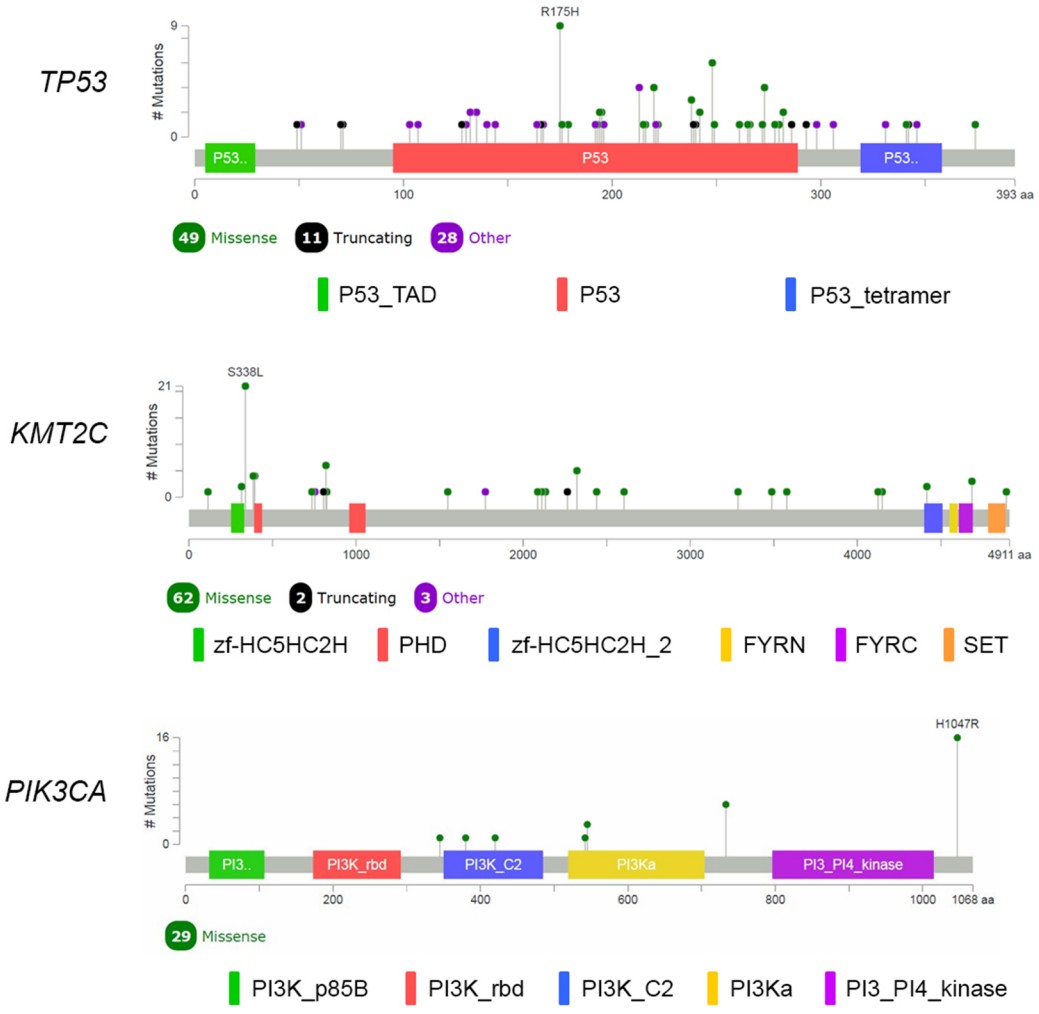

**Figure 3 Mutational spectrum of *TP53, KMT2C* and *PIK3CA* in TNBC.** The images showed protein domains and the positions of specific mutations with most common type of mutations in each genes labeled. A black dot indicated a truncating mutation; a green dot indicated a missense mutation; and a purple dot indicated other types of mutation. 

## DISCUSSION

Triple negative breast cancer is a heterogeneous disease with marked variation in clinical characteristics and response to treatment (*Blows et al., 2010*). Genome data from previous studies confirmed spectrum of mutational profiles in TNBC are diverse between each patients and cohorts (*Cancer Genome Atlas Network, 2012*; *Pereira et al., 2016*; *Shah et al., 2012*; *Stephens et al., 2012*). Though there are several explanations for diverse genome landscape in TNBC, patient's ethnicity could play a significant role in this discrepancy. For the first time, this study provided the mutation profile of TNBC from Thai breast cancer patients. The data from our cohort showed that, besides *TP53* which is the most frequently mutated gene in TNBC, Thai TNBC patients have much higher mutation frequencies in many cancer genes than Western patients. This increase could be due to its exceptionally different pattern of somatic genome alterations in

Thai TNBC or its representation of higher tumor mutation burden that occurs extensively throughout cancer genome or both.

TP53 remains the most commonly mutated gene in Thai TNBC similar to other reported studies (Lefebvre et al., 2016; Cancer Genome Atlas Network, 2012; Pereira et al., 2016). This gene is widely considered guardian of the genome due to its crucial function in maintaining genome integrity, regulating cell cycle and initiating apoptosis. Numerous types of mutations in TP53 are found in various cancers and mutations occur throughout the entire TP53 as expected in loss-of-function (LOF) mutations in tumor suppressor gene. We also identify several recurring mutations in TP53 including R175H, Y220C, R248Q and R273H, which together account for 26% of TP53 mutant group. These four mutations are hotspot mutations that believed to be oncogenic missense variants (Soussi & Wiman, 2015). Previous study in breast cancer showed that oncogenic TP53 variants in DNA binding domain (amino acid position 102-292) are associated with reduced survival compared to wild-type TP53 with an exception of Y220C, which is associated with better survival in breast cancer (Olivier et al., 2006). However, such association could not be identified likely due to very few mortality during our 3-year follow-up period.

PIK3CA mutations are usually enriched in hormonal receptor-positive tumor at 29–45%, with lower frequency in TNBC (Cancer Genome Atlas Network, 2012). PIK3CA is the most commonly mutated oncogene in Thai TNBC cohort similar to other breast cancer studies. Nevertheless, the mutation frequency of PIK3CA in Thai TNBC is much higher than previously published data (Cancer Genome Atlas Network, 2012; Pereira et al., 2016; Shah et al., 2012). H1074R is the major PIK3CA hotspot mutation followed by K733R and E545K. These variants are known oncogenic gain-of-function mutations found in multiple types of cancers. Together with other PI3K pathway members, TNBC harboring mutations in PIK3CA and its related genes could be potential targets for PI3K inhibitors.

BRCA1 and BRCA2 encode proteins that become parts of a complex that repairs double-strand DNA breaks. They are critical for maintaining genome integrity. Breast cancers occurred in most germline BRCA1 mutation carriers are TNBC. By contrast, there is no characteristic breast cancer subtype in BRCA2 carriers (Atchley et al., 2008). Most TNBC patients do not harbor germline BRCA1 or BRCA2 mutations. However, pathological high-grade breast cancers and TNBC often showed somatic mutations or abnormal expression of BRCA1 or BRCA2 (von Wahlde et al., 2017). Thai TNBC displayed high prevalence of somatic alterations in both genes. By comparison with three Western cohorts, the difference was less obvious because the mutation frequencies did vary from one cohort to another. Nevertheless, defects in genome repair machinery related to BRCA1 and BRCA2 mutations could make treatment with platinum-based chemotherapy or PARP inhibitors more effective in those patients (Papadimitriou, Mountzios & Papadimitriou, 2018; Vollebergh et al., 2014).

The KMT2C is noticeable because it is the second most commonly mutated gene (58%) in our Thai TNBC cohort. KMT2C is a gene in myeloid/lymphoid or MLL family and encodes MLL3 lysine-specific histone methyltransferase enzyme. H3 'Lys-4' methylation in histone by the enzyme represents a specific tag for epigenetic

transcriptional activation (*Rao & Dou, 2015*). LOF mutations in *KMT2C* are found in myeloid leukemia, melanoma, glioblastoma multiforme, hepatocellular carcinoma, esophageal cancer, colorectal cancer and pancreatic cancer (*Fujimoto et al., 2012*; *García-Sanz et al. 2017*; *Liu et al., 2015*; *Lu et al., 2016*; *Song et al., 2014*). Though the biological role of MLL3 histone methyltransferase in carcinogenesis remains unknown, LOF mutations and downregulation of this gene in cancers suggest that *KMT2C* may act as tumor suppressor gene (*Xia et al., 2015*). *KMT2C* mutations are also reported in TNBC (*Liu et al., 2015*). However, Thai TNBC demonstrates substantially higher prevalence of *KMT2C* mutations than TNBC data from Caucasian population (*Cancer Genome Atlas Network, 2012*; *Pereira et al., 2016*). A study from African American TNBC also displayed higher *KMT2C* mutations than Caucasian patients (*Ademuyiwa et al., 2017*). A recently published data identified *KMT2C* mutations in 21% of breast cancer patients from Singapore and Korea (*Yap et al., 2018*). Interestingly, *KMT2C* has one identified hotspot variant; S338L (31%) which has been previously reported in colorectal cancer (*Lu et al., 2016*). This observation suggests that epigenetic change may contribute to the development of TNBC and play significant role in Thai TNBC patients. This data could lead to a new insight on epigenetic role of breast carcinogenesis. Further investigation is warranted to provide better understanding on mechanisms of *KMT2C* and a novel treatment strategy.

As previously mentioned, Thai TNBC has generally higher mutation burden than breast cancer in Western patients. The reason for this finding remains not fully explained. Previous study showed that a subset of TNBC harbors somatic mutations in genome repair system (*Shah et al., 2012*; *Stephens et al., 2012*). Higher tumor mutational load was observed in hormonal receptor negative than hormonal receptor-positive breast cancer (*Barrett et al., 2018*; *Haricharan et al., 2014*). Early phase of anti-PD-1 clinical trials also showed higher response rate in TNBC than hormonal receptor-positive breast cancer (*Nanda et al., 2016*; *Rugo et al., 2018*). Our finding suggests that immunotherapy could provide benefit to some Thai TNBC patients.

In this study, the data from Thai TNBC would contribute much-needed information from Asian patients to breast cancer genome landscape and provide another evidence on role of KMT2C in breast carcinogenesis. However, our study has three major limitations. First, only patients from Siriraj and King Chulalongkorn Memorial Hospitals were recruited. Both Bangkok-based university hospitals were the largest hospitals in Thailand and served as major referral centers of Thailand's healthcare system. Though breast cancer patients treated at our hospitals came from all over the country, many patients from other regions of Thailand who could be treated locally did not participate in the study. The data may not represent the whole picture of Thai TNBC patients. Second, the follow-up period up to 3 years was not long enough to observe clinically meaningful association between genomic alterations and clinical outcomes. Further study would be required. Third, genome sequencing was done on only tumor gDNA extracted from FFPE samples. It has been recognized that the quality of gDNA from FFPE is lower than fresh samples and potentially causes variant call discrepancy. Our study chose to focus on list of cancer-associated genes and apply variant call only when genomic regions have sufficient sequencing depth.

This approach has shown to minimize erroneous variant calls, improve precision and acceptable correlation with matched normal-tumor pair sequencing (*Chalmers et al., 2017*; *Oh et al., 2015*; *Teer et al., 2017*). Nevertheless, comparison of our TMB data with previously published studies could be limited by differences in study designs and data analysis methods.

## CONCLUSION

In conclusion, this study is the first cohort of Thai TNBC patients that demonstrated a distinctive genome alterations including higher mutational burden, higher mutation frequencies on several cancer-associated genes and mutations in *KMT2C*. These results support the genomic heterogeneity between Caucasian and Thai TNBC and could present the new therapeutic approach on histone modification and immunotherapy in TNBC patients. Further investigation is warranted to provide better understanding on role of KMT2C in breast carcinogenesis.

## ABBREVIATIONS

| | |
|---|---|
| **BMI** | Body mass index |
| **CADD** | Combined annotation dependent depletion |
| **COSMIC** | Catalogue of somatic mutations in cancer |
| **DNA** | Deoxyribonucleic acid |
| **ER** | Estrogen receptor |
| **ExAC** | Exome aggregation database |
| **FFPE** | Formalin-fixed, paraffin-embedded |
| **gDNA** | Genomic deoxyribonucleic acid |
| **HER2** | Human epidermal growth factor receptor 2 |
| **HGMD** | Human gene mutation database |
| **LOF** | Loss of function |
| **LRT** | Likelihood ratio test |
| **OMIM** | Online Mendelian inheritance in men |
| **PAM50** | Prosigna intrinsic subtype of breast cancer |
| **PARP** | Poly (ADP-ribose) polymerase |
| **PI3K** | Phosphatidylinositide 3-kinases |
| **PR** | Progesterone receptor |
| **SIFT** | Sorting intolerance from tolerance |
| **SNP** | Single nucleotide polymorphism |
| **SO** | Sequence ontology |
| **TCGA** | The cancer genome atlas |
| **TNBC** | Triple negative breast cancer |
| **UTR** | Untranslated region |

## ACKNOWLEDGEMENTS

We wish to thank all participants for their cooperation and contribution to our study. We thank all physicians and health professionals for their patient's clinical care. We also thank the Research University Network (RUN)—Thailand for the administrative support.

### Funding

This work was supported by the National Research Council of Thailand (Grand Challenge Grant) to Chanin Limwongse, Bhoom Suktitipat and Manop Pithukpakorn; Siriraj Research and Development Grant (Grant R015934003) to Suvimol Niyomnaitham; Siriraj Core Research Facility (SiCRF) Grant to Manop Pithukpakorn; Siriraj Chalermphrakiat Grant to Suvimol Niyomnaitham, Bhoom Suktitipat, Norasate Samarnthai, Wanna Thongnoppakhun, Chanin Limwongse and Manop Pithukpakorn; the Crown Property Bureau Foundation Grant to Bhoom Suktitipat; Thanapat Fund (D003752) to Manop Pithukpakorn. The funders had no role in study design, data collection and analysis, decision to publish or preparation of the manuscript.

### Grant Disclosures

The following grant information was disclosed by the authors:
National Research Council of Thailand: Grand Challenge Grant.
Siriraj Research and Development Grant: R015934003.
Siriraj Core Research Facility (SiCRF).
Siriraj Chalermphrakiat Grant.
Crown Property Bureau Foundation.
Thanapat Fund: D003752.

### Competing Interests

The authors declare that they have no competing interests.

### Author Contributions

- Suvimol Niyomnaitham conceived and designed the experiments, performed the experiments, analyzed the data, contributed reagents/materials/analysis tools, prepared figures and/or tables, authored or reviewed drafts of the paper, approved the final draft.
- Napa Parinyanitikul conceived and designed the experiments, performed the experiments, analyzed the data, contributed reagents/materials/analysis tools, authored or reviewed drafts of the paper, approved the final draft.
- Ekkapong Roothumnong conceived and designed the experiments, performed the experiments, analyzed the data, contributed reagents/materials/analysis tools, prepared figures and/or tables, approved the final draft.
- Worapoj Jinda performed the experiments, analyzed the data, contributed reagents/materials/analysis tools, prepared figures and/or tables, approved the final draft.
- Norasate Samarnthai performed the experiments, contributed reagents/materials/analysis tools, approved the final draft.
- Taywin Atikankul performed the experiments, contributed reagents/materials/analysis tools, approved the final draft.
- Bhoom Suktitipat conceived and designed the experiments, analyzed the data, contributed reagents/materials/analysis tools, authored or reviewed drafts of the paper, approved the final draft.
- Wanna Thongnoppakhun conceived and designed the experiments, performed the experiments, analyzed the data, contributed reagents/materials/analysis tools, prepared figures and/or tables, authored or reviewed drafts of the paper, approved the final draft.
- Chanin Limwongse conceived and designed the experiments, analyzed the data, contributed reagents/materials/analysis tools, authored or reviewed drafts of the paper, approved the final draft.
- Manop Pithukpakorn conceived and designed the experiments, analyzed the data, contributed reagents/materials/analysis tools, prepared figures and/or tables, authored or reviewed drafts of the paper, approved the final draft.

### Human Ethics

The following information was supplied relating to ethical approvals (i.e., approving body and any reference numbers):

The study protocol was approved by the Siriraj and King Chulalongkorn Memorial Hospital Institutional Review Boards (Protocol No. 175/2559 and 642/2557).

### Data Availability

The following information was supplied regarding the deposition of DNA sequences:

The sequence data of this study are accessible via Figshare: Suktitipat, Bhoom; Pithukpakorn, Manop (2018): Triple Negative Breast Cancer Data. figshare. Fileset. DOI 10.6084/m9.figshare.7467401.v2

NCBI BioProject database, ID PRJNA514873.

### Supplemental Information

Supplemental information for this article can be found online at http://dx.doi.org/10.7717/peerj.6501#supplemental-information.

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
