# Peer review of "Tumor mutational profile of triple negative breast cancer patients in Thailand revealed distinctive genetic alteration in chromatin remodeling gene"

_PeerJ, doi:10.7717/peerj.6501_

## Round 0.1 · original submission · Major Revisions

The reviewers has some remarks demanding revision (rather minor). Please consider all the comments. Waiting your updates.

Welcome to submit updated manuscript earlier.

Happy coming New Year!

Reviewer 1 ·

Basic reporting

No comment.

Experimental design

It is not clear which material was used in the study: primary tumor tissues and/or metastatic lymph nodes?

It appears that lines 173-178 should be transferred to the section Study Population.

Validity of the findings

The study result showed that Thai TNBC has higher tumor mutation burden than previously reported data (ref. 10-12). But the authors compare their results with data from three other studies that have a different design. The analysis in those studies was carried out with the involvement of not only tumor, but also healthy tissue (matched tumour/normal pairs). In this way, germ-line genetic variants could be distinguished from somatic mutations and excluded from analysis. Moreover, in the at least one study (ref. 12) the frozen tumor material was studied. In the other two articles, unfortunately, data on the material is not presented. Formalin fixation leads to significant changes in DNA. Do the authors believe that such differences in design could lead to a higher tumor mutation burden?

I can’t fully agree with the statement lines 223-226. A large number of examples have been described when synonymous mutations are significant for carcinogenesis and lead to the synthesis of mutant proteins. So they can create neoantigens and trigger immune response too. See Cell. 2014 Mar 13;156(6):1324-1335. and EMBO Mol Med. 2016 May 2;8(5):442-57.

Additional comments

Line 156: It is necessary to clarify the abbreviation.

Reviewer 2 ·

Basic reporting

Structure article and English language conforms to requirements

Experimental design

Experimental design is standard for this kind of research

Validity of the findings

The results are consistent with previously published data and complement them with new ones. The limitations of study have been described honestly.

Additional comments

The surgically removing primary breast tumors and lymph nodes is standard treatment described in the "Methods" (lines 112). In order to avoid misunderstanding I recommend the authors to describe in more detail which samples were analyzed (primary tumors or lymph node metastases).

---

## Round 0.2 · accepted · Accept

Both reviewers recommend accept the manuscript in current form, good job. I believe this topic on tumor profiling and possible breast cancer treatment is very important out of Thailand too, encourage you and co-authors continue the study.

# Reviewer 1 ·

Basic reporting

No comment.

Experimental design

No comment.

Validity of the findings

No comment.

Additional comments

Thank you.
I am satisfied with the answers and agree with the corrections.

Reviewer 2 ·

Basic reporting

Structure article and English language conforms to requirements

Experimental design

Experimental design is standard for this kind of research

Validity of the findings

The results are consistent with previously published data and complement them with new ones. The limitations of study have been described honestly.

Additional comments

Comments taken into account. The manuscript made all the necessary amendments. Article recommended for printing